# Tumor infiltrating lymphocytes associated competitive endogenous RNA networks as predictors of outcome in hepatic carcinoma based on WGCNA analysis

**Ying He[1], Rui Xu[1], Li Peng[1], Xiaoyu Hu◉[2]***

**1** Department of College of Clinical Medicine, Chengdu University of Traditional Chinese Medicine, Chengdu, Sichuan, China, **2** Department of Infectious Disease, Hospital of Chengdu University of Traditional Chinese Medicine, Chengdu, Sichuan, China

* xiaoyuhu@aliyun.com

**Data Availability Statement:** All relevant data are available within the paper and its Supporting Information files.

## Abstract

### Background

The important regulatory role of competitive endogenous RNAs (ceRNAs) in hepatocellular carcinoma (HCC) has been confirmed. Tumor infiltrating lymphocytes (TILs) are of great significance to tumor outcome and prognosis. This study will systematically analyze the key factors affecting the prognosis of HCC from the perspective of ceRNA and TILs.

### Methods

The Cancer Genome Atlas (TCGA) database was used for transcriptome data acquisition of HCC. Through the analysis of the Weighted Gene Co-expression Network Analysis (WCGNA), the two modules for co-expression of the disease were determined, and a ceRNA network was constructed. We used Cox regression and LASSO regression analysis to screen prognostic factors and constructed a risk score model. The Gene Expression Omnibus (GEO) was used to validate the model. The Kyoto Encyclopedia of Genes and Genomes (KEGG) was used for mRNAs functional analysis. The cell composition of TILs was analyzed by the CIBERSORT algorithm, and Pearson correlation analysis was utilized to explore the correlation between TILs and prognostic factors.

### Results

We constructed a ceRNA regulatory network composed of 67 nodes through WGCNA, including 44 DElncRNAs, 19 DEGs, and 4 DEmiRNAs. And based on the expression of 4 DEGs in this network (RRM2, LDLR, TXNIP, and KIF23), a prognostic model of HCC with good specificity and sensitivity was developed. CIBERSORT analyzed the composition of TILs in HCC tumor tissues. Correlation analysis showed that RRM2 is significantly correlated with T cells CD4 memory activated, T cells CD4 memory resting, T cells CD8, and T cells follicular helper, and TXNIP is negatively correlated with B cells memory.

**Funding:** The author(s) received no specific funding for this work.

**Competing interests:** The authors have declared that no competing interests exist.

## Conclusions

In this study, a ceRNA with prognostic value in HCC was created, and a prognostic risk model for HCC was constructed based on it. This risk score model is closely related to TILs and is expected to become a potential therapeutic target and a new predictive indicator.

## Introduction

Liver cancer is classified into hepatocellular carcinoma (HCC) and intrahepatic cholangiocarcinoma [1]. HCC accounts for 90% of primary liver cancers with estimated deaths ranking fourth among all kinds of cancers [2]. HCC is characterized by atypical clinical symptoms, difficult early diagnosis, a high degree of malignancy, early metastasis, and poor efficacy [3–5]. The incidence of liver cancer is still increasing, and the survival rate of 5-year is less than 18% [6]. Therefore, it is of great importance to understand the molecular mechanism of HCC development and to study potential foreseeable biological indicators to improve patient prognosis. The development of high throughput sequencing and computer technologies provides a methodological approach for the identification of disease-related genes in HCC, and also provides a new perspective on the study of cancer.

In recent years, the view that ceRNA is an important way to regulate the progression of various cancers including HCC has been widely accepted [7]. In the ceRNA network, microRNA (miRNA) is the core element, and lncRNA can produce sponge-like effects compete with ceRNA for miRNA response elements, thereby regulating mRNA expression, which is the so-called ceRNA hypothesis proposed by Salmena et al [8]. As a non-coding single-stranded RNA, miRNA is composed of approximately 22 nucleotides, and its binding site is located in the 3'-untranslated region, which can induce degradation of target mRNA, thereby inhibiting gene expression at the post-translated level and playing a variety of biological effects [9,10]. Regulating the expression of miRNA can affect a variety of cell functions. For example, loss of miRNA-10a (miR-10a) can induce intestinal tumors, and miRNA-138 can inhibit HCC cells by inhibiting SOX9 [11,12]. The ceRNA network is a major mode of regulation of miRNA expression. In gastric cancer, lncRNA HOTAIR can act as ceRNA, adsorb miR-331-3p, which in turn affects the proliferation, migration, and invasion of gastric cancer cells [13]. Therefore, we have reason to believe that ceRNA also plays an equally crucial regulatory role in HCC. Based on this, we will screen out the differentially expressed RNA in HCC tissues through the TCGA database to construct a ceRNA network, and explore the biological function and prognostic value of this network in HCC.

TILs are lymphocytes that infiltrate the stroma of tumor cells, and activated TILs can induce tumor cell apoptosis [14]. TILs have been proved to be associated with the survival of cancer patients. For example, the number of B cells infiltrating tumor tissues is believed to be correlated with the survival rate of a variety of cancer patients including breast cancer, ovarian cancer, and melanoma [15,16]. Meanwhile, TILs seem to have a predictive effect on neoadjuvant chemotherapy response [17]. At present, cancer immunotherapies are a key component of cancer clinical treatment [18]. As an immune organ, the liver contains a great number of congenital and adaptive immune cells [19]. The liver has been confirmed to contain numerous CD69 + NK cells [20]. Therefore, in-depth study of immune cells infiltrating in HCC tissues to find potentially related molecules that may reflect or regulate is essential for developing new drugs, enhancing the effectiveness of HCC immunotherapy, and predicting the prognosis of HCC patients. CIBERSORT is considered to be a reliable tool for calculating immune cell

composition with excellent performance. Its main advantage lies in high sensitivity and specificity. It can quantify 22 human immune cell phenotypes at the same time, and through characterizing ~ 500 marker genes to calculate the relative ratio of each cell type [21]. WGCNA uses hierarchical clustering to identify clusters (modules) of highly related genes, uses module characteristic genes to define and classify clusters, and associates the modules with each other and the traits of external samples [22]. It is the most commonly used method for constructing co-expression networks and has been widely used to predict gene function, detect gene mutations in cancer, and discover new disease biomarkers [23,24]. Based on this, this study will construct a co-expression network through WGCNA, screen key genes in the ceRNA, and evaluate TILs in liver tumor tissues by CIBERSORT. In addition, the correlation between candidate genes in the constructed network and TILs in HCC tissues and their prognostic value was investigated.

## Materials and methods

### Data access

We obtained high-throughput RNA-sequencing data from the TCGA database (https://portal.gdc.cancer.gov/), including transcript data of 424 HCC patients and 425 patients miRNA-sequencing data. To eliminate the influence of gene length and sequencing depth on gene expression level, we calculated the reads per kilobase per million mapped reads (rpkm) for subsequent analysis. The ensemble grch38.84 was used for gene symbol conversion and annotate RNA categories. External ChIP-seq data sets for validation were downloaded from GEO database (https://www.ncbi.nlm.nih.gov/geo) including GSE14520, GSE22058, GSE25097, GSE57957 and GSE76427.

### Identification of differentially expressed RNAs

Differentially expressed genes (DEGs), lncRNAs (DElncRNAs), and miRNA (DEmiRNAs) were performed by the "limma" Bioconductor R package [25]. The significance thresholds were identified of | log2 fold change (FC) | > 1 and false discovery rate (FDR) < 0.05. Finally, the volcano map was drawn using the "ggplot2" package in R software, while the heatmap was drawn using the "pheatmap" R package.

### Construction of mRNA co-expression network

Based on mRNA expression, we used the "WCGNA" R package (Version 1.69) to construct the co-expression networks and screened the disease-related core modules [22]. The data set was grouped according to whether the samples were tumor tissues or not, including 374 tumor tissues and 50 non-tumor tissues. The nodes in the network represent mRNAs that are connected when the Pearson correlation coefficient between these nodes is greater than a specific threshold. In this study, the R-square cut-off value was set at 0.9. Conversion of a correlation matrix to adjacency matrix requires parameter β, and the value of β is set in such a way that the resulting adjacency matrix needs to approximate the scale-free topology feature. In this study, the soft threshold power is the parameter β, which is determined to be 8. Next, the topological overlap matrix (TOM) was calculated based on the adjacency matrix, and 1-TOM was used as a distance metric to perform the average linkage hierarchical clustering [26]. Correlation between module eigengenes and traits was evaluated with Pearson's r correlation. The resulting co-expression network was built with a minimum module size of 50, merge cut height of 0.25, and signed network type.

## Function annotation of mRNAs

Metascape (http://metascape.org) was used for KEGG analysis of mRNAs in two key modules, which showed the highest levels of positive or negative correlation with HCC, to understand the key pathways in the pathological process of HCC. In addition, the mRNAs in the ceRNA network we constructed also used this method for pathway analysis to explore the function of this ceRNA.

## Construction of the DElncRNA-DEmiRNA-DEG regulatory network

We first used the miRcode database (https://www.mircode.org/) to forecast the miRNAs that interact with DElncRNAs, and cross the predicted miRNAs with DEmiRNAs to obtain the DElncRNA-DEmiRNA interactions pairs in the target dataset. Then we used TargetScan database (www.targetscan.org), miRDB database (http://mirdb.org/miRDB/), and miRTarBase database (http://mirtarbase.mbc.nctu.edu.tw) to predict the target mRNAs of the DEmiRNAs screened in the previous step, only predicted mRNAs with consistent predictions from the three databases were selected for further analysis. The predicted mRNAs were intersected with DEGs and the mRNAs in the core modules screened by WGCNA, and the final intersectant mRNAs were used for the construction of ceRNAs (Fig 4B). Cytoscape V3.7.1. was used for visualization.

## Identification of prognostic signatures in ceRNA network

The 19 mRNAs in ceRNAs were further used to construct a prognostic risk score model. To make the model more reproducible, we also performed differential expression analysis on GSE14520, and selected mRNAs that were differentially expressed in GSE14520 and consistent with the trend of changes in the expression of these 19 mRNAs for the final analysis. Firstly, the 19 factors were assessed using the univariate Cox proportional hazard regression model, then the least absolute shrinkage and selection operator (LASSO) was applied to carry out dimensionality reduction for the variables in the regression model, and finally, a multivariate Cox regression analysis was carried out [27,28]. The "glmnet" in R software was used for LASSO Cox variable selection and model building, and the "survival" R package was used for comparison of the survival curves. The comprehensive risk score of each patient was calculated according to the linear combination of expressed values weighted by the LASSO Cox regression coefficient, and the formula is as follows.

$$\text{Risk score} = \text{Exp1} * \beta1 + \text{Exp2} * \beta2 + \text{Exp3} * \beta3 + \ldots \ldots \text{Expi} * \beta\text{i}.$$

In this equation, Exp represents the value of key mRNAs, and β represents the corresponding coefficient of the multivariate Cox regression analysis. According to the prognostic risk score, the HCC patients were regrouped. Cases with RiskScore > median were defined as the high-risk group, and vice versa, the low-risk group. We used "survplot" R package to plot the Kaplan-Meier (KM) survival curves In the high and low-risk groups of patients, and the Log-rank was used to compare the overall survival (OS) of the inter-group. We also conducted survival analysis on all signatures that make up the ceRNA network. The specificity and sensitivity of the model in prognosis prediction were evaluated according to the areas under the ROC curve at 1, 3, and 5 years.

## Prognosis and expression validation for key DEGs

The Human Protein Atlas Database (HPA) (http://www.proteinatlas.org) is used to obtain representative immunohistochemical images of key DEGs between normal and hepatic

carcinoma tissue. GSE14520, GSE22058, GSE25097, GSE57957, and GSE76427 are all used for expression verification of key prognostic genes. The GSE14520 data set contains 225 HCC tumors and 220 adjacent paracancerous tissues, with 221 complete survival data, this data set will be used as a validation set to verify the accuracy and sensitivity of the prognostic risk score model.

## Analysis of lymphocyte infiltration through CIBERSORT

For analyzing the immunobiological characteristics of HCC patients, we use CIBERSORT (http://cibersort.stanford.edu/) to classify and quantify the abundance of TILs. The input data is the RNA expression matrix downloaded directly from the TCGA database [29]. For the next analysis, only cases with CIBERSORT P-value < 0.05 can be used. Violin plots were generated using the "vioplot" R package. Meanwhile, Spearman correlation analysis was used to investigate the correlation between prognosis-related DEGs and infiltrating lymphocytes.

## Statistical analyses

Data were plotted using GraphPad Prism 5.0 and analyzed using SPASS Statistics 2.0. The nonparametric Mann-Whitney-U test was adopted to test the differences between groups. The level of significance was set at $\alpha = 0.05$.

# Results

## Differential expression analysis

The DEGs were screened out using FDR < 0.05 and $|\log2FC| > 1$ as the cutoffs. A total of 2769 mRNAs (1741 downregulated and 1028 upregulated) (Fig 1A), 259 lncRNAs (119 downregulated and 140 upregulated) (Fig 1B), and 127 miRNAs (58 downregulated and 69 upregulated) (Fig 1C) were differently expressed between the HCC tumor and normal liver tissues. Differential analysis of validation set GSE14520 showed that there were 9405 differentially expressed genes, of which 4879 genes were up-regulated and 4526 genes were down-regulated (Fig 1D).

## Construction of WGCNA and identification of hub modules

We used the "WGCNA" R package to explore the expression profiles of mRNAs in 374 HCC tissues and 50 normal tissues. The mRNAs were hierarchically clustered with the cut line value set to 20,000 to remove outliers (Fig 2A), but no samples were removed in this study. The independence and average connectivity of the modules is determined by the power value.

Setting the power value to 8 can make the scale-free $R^2$ reach 0.9, which is the lowest threshold (Fig 2B), to better reflect the scale-free topology of the co-expression network. According to the topological overlap measure (TOM), a total of 7 color modules are determined through dynamic tree cutting. The mRNAs that could not be included in any module were put into the gray module (Fig 2C). The correlation between modules and traits is shown in Fig 2D, in which magenta module (r = 0.86, P < 0.01) and yellow module (r = -0.57, P < 0.01) represent the most positive and negative correlations with HCC. The number of mRNAs contained in these two modules is 214 (magenta module) and 623 (yellow module) (Fig 2E and 2F), and all of these 837 mRNAs identified by WGCNA will be used in the next analysis.

The mRNAs in the selected core modules were performed KEGG pathway analysis. The pathway analysis of 214 mRNAs involved in the magenta module showed that they were mainly rich in mineral absorption (ko04978), malaria (ko05144), and MAPK signaling pathways (hsa04010), while the 623 mRNAs in the yellow module were mainly enriched in the cell

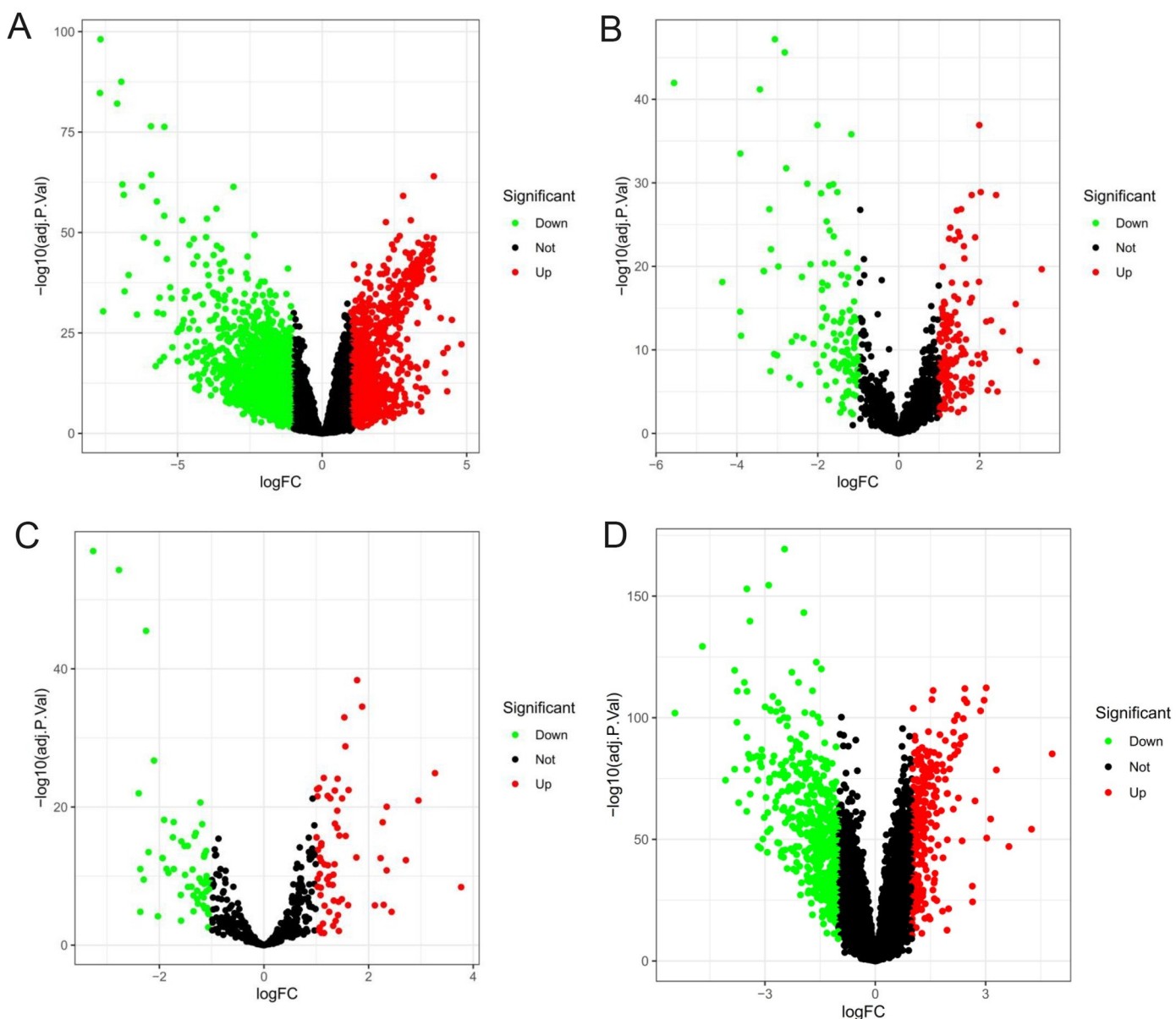

**Fig 1. Volcanic diagram of differentially expressed RNAs in patients with HCC.** (A) mRNA. (B) lncRNA. (C) miRNA. (D) Differential analysis of GSE14520.

cycle (hsa04110), DNA Replication (ko03030) and p53 signaling pathway (ko04115), etc (Fig 3A and 3B).

## Construction of the ceRNA network

Firstly, the miRNAs targeted by 259 DElncRNAs were foreseen using miRcode database and the predicted miRNAs were mapped to DEmiRNAs. A total of 175 DElncRNAs-DEmiRNAs pairs were identified, which included 58 DElncRNAs and 9 DEmiRNAs. Secondly, we used the 9 DEmiRNAs selected in the previous step to predict their target mRNAs through miRDB, miRTarBase, and TargetScan databases. The predicted mRNAs were intersected with DEGs and the mRNAs in the core modules screened by WGCNA, and 19 DEmiRNAs-DEGs pairs were obtained. Finally, we constructed a ceRNA network through Cytoscape including 44

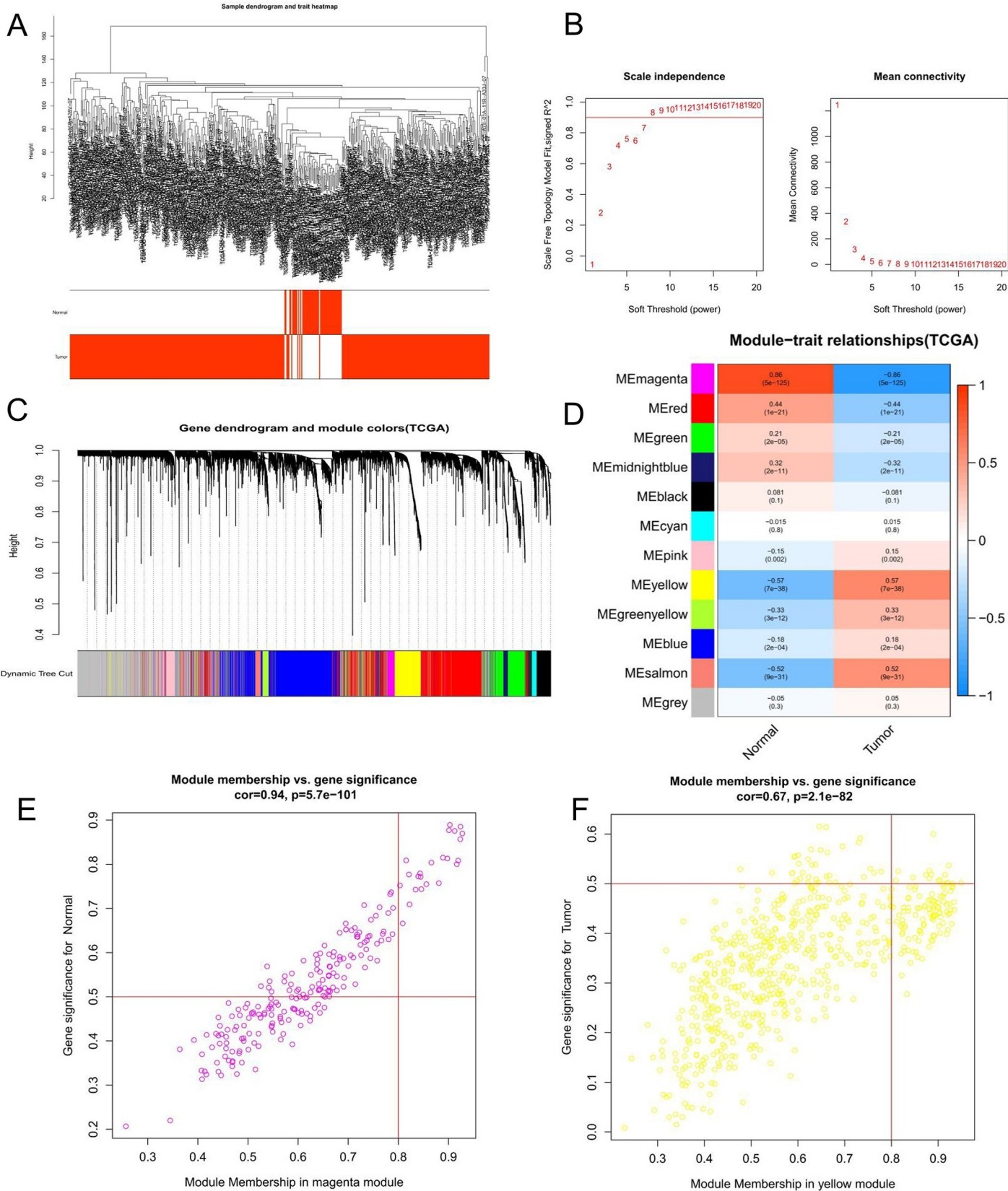

**Fig 2. Weighted correlation network analysis for the mRNAs.** (A) Hierarchical clustering tree of 374 tumor and 50 normal liver tissues mRNAs expression patterns. (B) Identification of power value. The red line represents $R^2 > 0.9$ when the power value is 8. (C) Clustering dendrogram and merging of mRNAs co-expression modules. (D) Correlation heatmap of mRNA modules and clinical traits. The correlation is related to color changes, red and blue represent positive and negative correlation, respectively. (E) Scatter diagram of mRNAs in the magenta module, which is positively correlated with HCC. (F) Scatter diagram of mRNAs in the yellow module, which is positively correlated with HCC.

DElncRNAs, 19 DEGs, and 4 DEmiRNAs (Fig 4A). The related flow chart is shown in Fig 4B. KEGG analysis shows that the pathway is mainly enriched in Hepatitis B (hsa05161) and Cell cycle (ko04110) (Fig 4B).

## Prognostic value of ceRNA network

We intersected the 19 DEGs included in the ceRNA network with the mRNAs in GSE14520 with statistically significant differences between tumor and normal tissues, and finally obtained

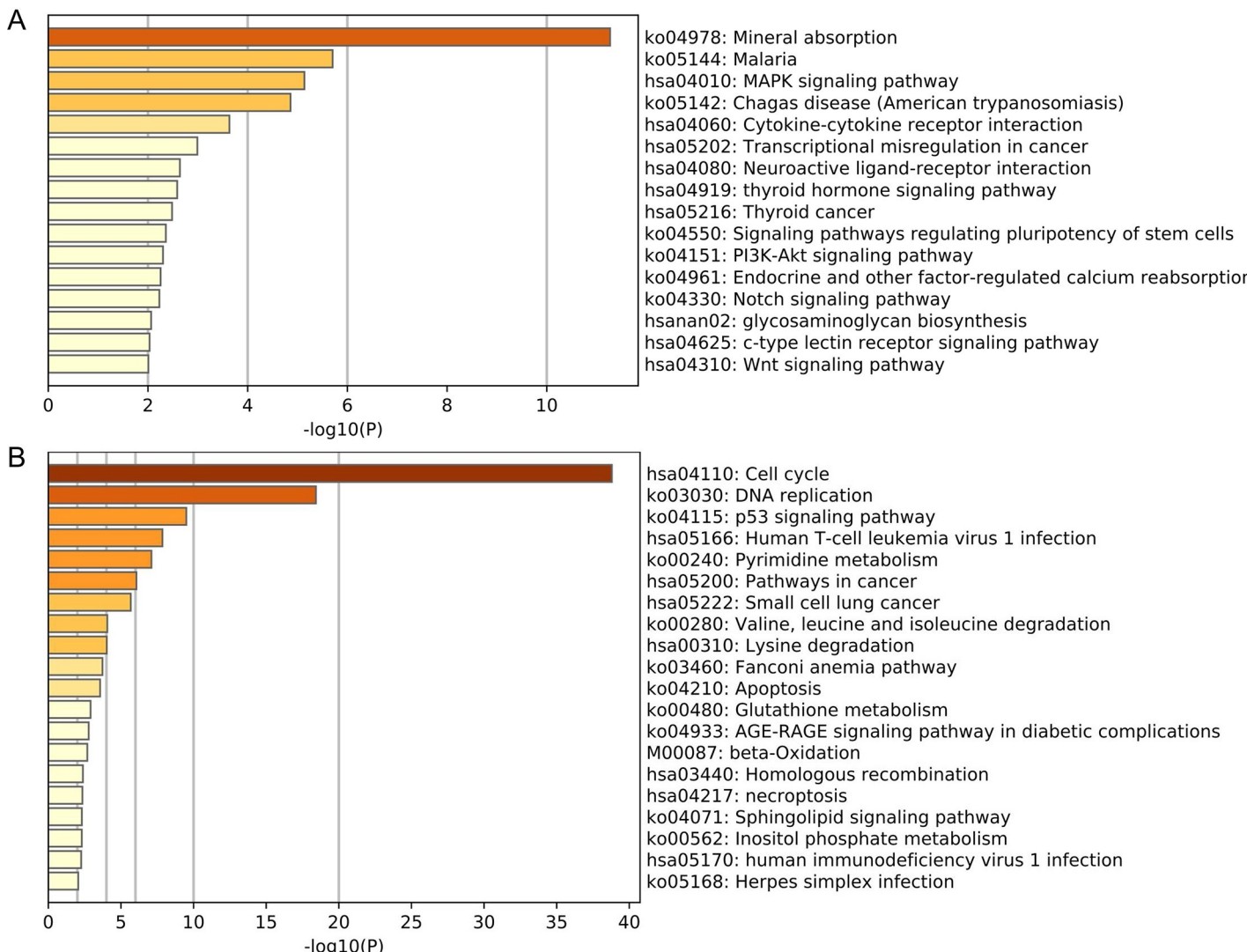

**Fig 3. The core module mRNAs enrichment analysis.** (A) The KEGG analysis of mRNAs involved in the magenta module. (B) The KEGG analysis of mRNAs involved in the yellow module.

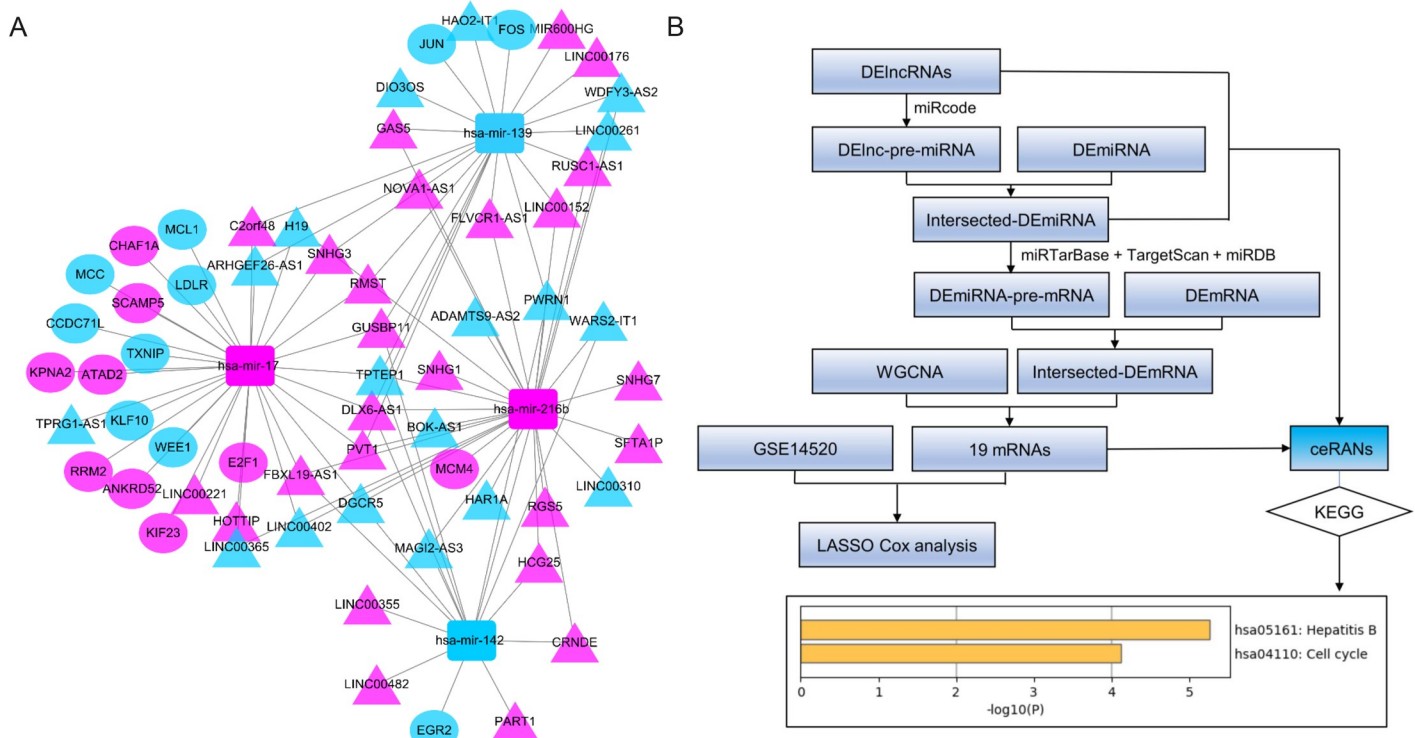

**Fig 4. The DEGs-DEmiRNAs-DElncRNAs network.** (A) ceRNA network. (B) Flow chart of construction and analysis of ceRNA network. Rectangles indicate DEmiRNAs, triangles indicate DElncRNAs, and ellipses indicate DEGs. There are blue and red marks that represent down-regulated and up-regulated RNAs, respectively. The KEGG analysis of the 19 mRNAs in the network was performed by the Metascape database with a P-value cutoff of 0.01.

a total of 14 mRNAs with the same trend of change (Fig 5A). These 14 mRNAs were used for Univariate Cox analysis to identify potential risk factors, and only 6 mRNAs with a statistical difference were selected (Table 1). Four mRNAs that remained after LASSO regression analysis were selected for the construction of the multivariate Cox prediction model (Fig 5B). In the prognosis model, four molecules with statistical differences are RRM2, LDLR, TXNIP, and KIF23 (Fig 5C). We calculated the risk score of each patient as follows:

$$\text{Risk score} = (0.019) * \text{Exp RRM2} + (-0.025) * \text{Exp LDLR} + (-0.003) * \text{Exp TXNIP} + (0.185) * \text{Exp KIF23}.$$

Patients were segmented into high-risk and low-risk groups using the median as a cut-off value. We used time-dependent ROC curves at 1, 3, and 5 years and KM plots to evaluate the three-mRNA signature in predicting the outcome of the HCC patients. In the training set, the area under the curve (AUC) for 1, 3, and 5 years of survival were 0.685, 0.668, and 0.643, respectively, which indicates that our prediction model constructed using these 4 mRNAs can accurately predict the survival of patients (Fig 5D). KM curve between high and low-risk groups showed a significant difference in OS (log-rank test P < 0.001) (Fig 5E). This result was further confirmed in the external validation set GSE14520. In the validation set, the AUC for 1, 3, and 5 years of survival were 0.616, 0.626, and 0.639, respectively. At the same time, KM curve also confirmed that the high-risk score group has a worse survival (Fig 5F and 5G).

To further explore the prognostic value of the ceRNA network we constructed, we also performed survival analysis on all the constituent elements of ceRNA. The results showed that 8

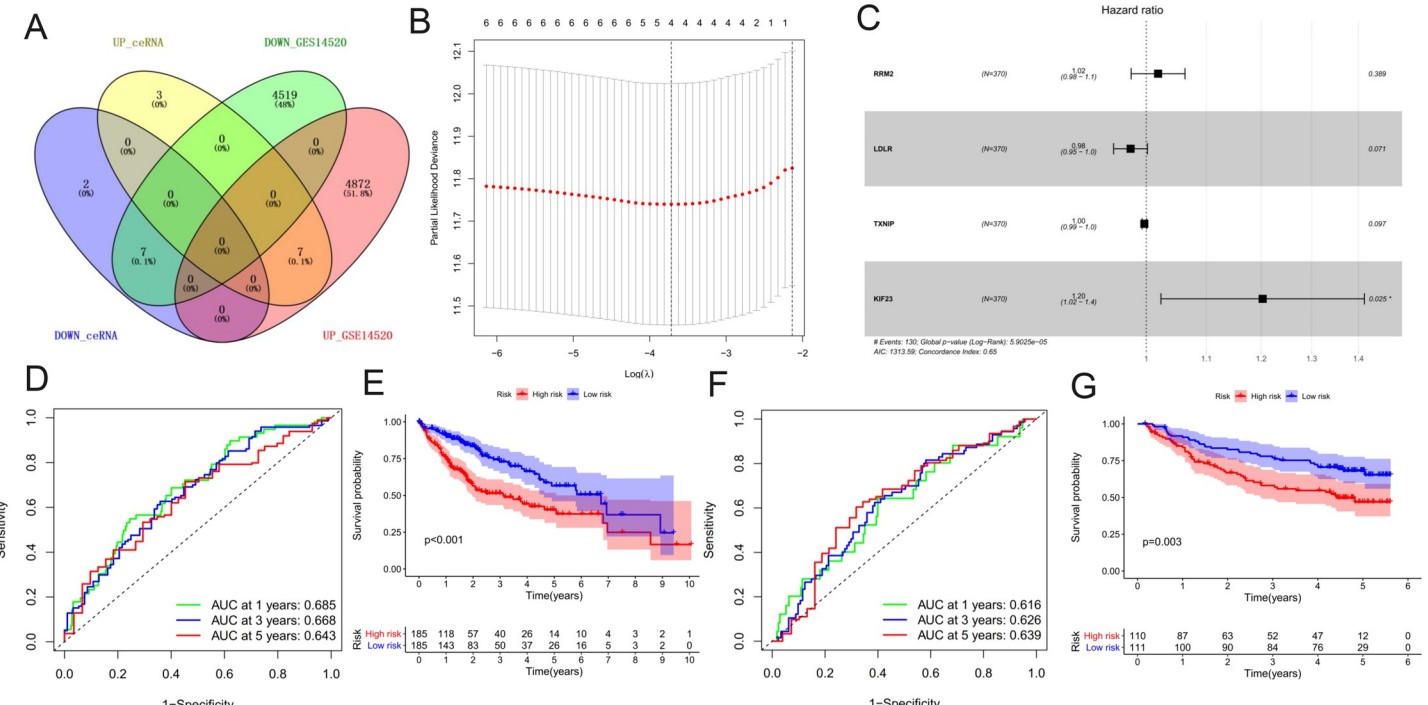

**Fig 5. Identification and evaluation of the 19 mRNAs signature.** (A) The 14 mRNAs used for the construction of the risk score model were produced by the intersection of DEGs in ceRNA and GSE14520. (B) Partial likelihood deviance of minimum number corresponds to the covariates revealed by the LASSO regression model. (C) Forest plots of hazard ratios (HR) of the four molecules with statistical differences OS-associated mRNAs. (D) The 1, 3, and 5 years of survival were analyzed by ROC curve to evaluate the efficiency of the clinical predictive model. (E) Survival curve of high and low-risk groups stratified by the prediction model. (F) The ROC curves of the external validation set GSE14520. (G) Survival curve of the external validation set GSE14520.

DEGs (CCDC71L, KPNA2, RRM2, E2F1, MCM4, CHAF1A, TXNIP, and KIF23), 10 DElncR-NAs (C2orf48, LINC00152, HAO2-IT1, SNHG3, DIO3OS, LINC00261, WARS2-IT1, FBXL19-IT1), and one DEmiRNA (miR-139-5p) were found to have statistical significance in predicting OS (Fig 6).

## Gene expression validation

We selected 5 data sets from the GEO database for external verification of 4 key model genes (RRM2, LDLR, TXNIP, and KIF23) expressions. The results showed that the expression of RRM2 and KIF23 was significantly up-regulated in HCC tumor tissues compared with adjacent tissues (P < 0.001). TXNIP was down-regulated in tumor tissues, with statistically significant differences in GSE14520, GSE25097, and GSE76427 (P < 0.001), while there was no

**Table 1. The Cox regression analysis of key mRNAs.**

| Factors | Univariate Cox regression | | Multivariate Cox regression | |
|---------|---------------------------|-----|-----------------------------|-----|
| | HR(95%CI) | P | HR(95%CI) | P |
| RRM2 | 1.058 (1.026–1.090) | < 0.001 | 1.019 (0.976–1.063) | 0.389 |
| E2F1 | 1.015 (1.003–1.026) | 0.016 | | |
| MCM4 | 1.050 (1.018–1.082) | 0.002 | | |
| LDLR | 0.965 (0.939–0.992) | 0.011 | 0.976 (0.950–1.002) | 0.071 |
| TXNIP | 0.996 (0.992–0.999) | 0.014 | 0.997 (0.993–1.001) | 0.097 |
| KIF23 | 1.302 (1.163–1.458) | < 0.001 | 1.203 (1.024–1.413) | 0.025 |

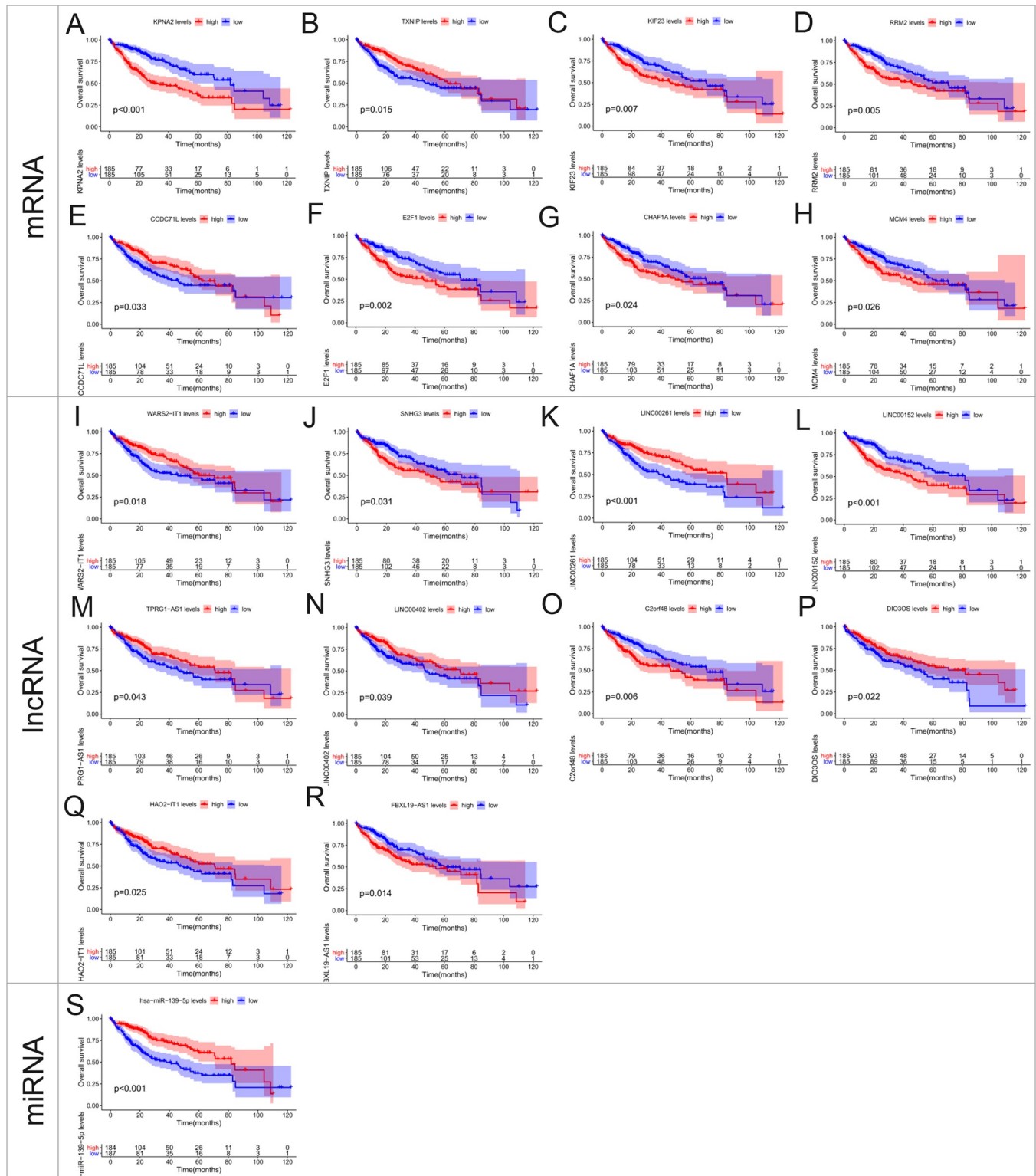

**Fig 6. Kaplan-Meier survival curves for all signatures on the ceRNA network we constructed.**

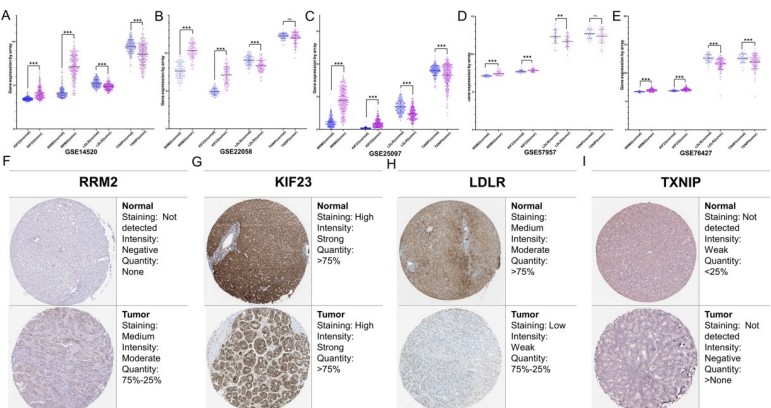

**Fig 7. Gene expression analysis validation.** External validation data set downloaded from GEO including (A) GSE14520, (B) GSE22058, (C) GSE25097, (D) GSE57957, and (E) GSE76427 used to compare the differential expression of the target genes between HCC and para-cancerous tissues. The immunohistochemical images from the HPA database showed the expression of (F) RRM2, (G) KIF23, (H) LDLR, and (I) TXNIP in HCC tumor and normal tissue.

statistically significant difference in GSE22058 and GSE57957 (P > 0.05). LDLR was confirmed to be down-regulated in HCC tumor tissues in all external data sets (P < 0.01) (Fig 7A–7E).

To assess the protein expression levels of these 4 mRNAs, we used the HPA database to obtain immunohistochemical images of normal and HCC tumor tissues. Among them, the difference of RRM2 was the most significant, the expression level of RRM2 in tumor tissues was significantly higher than that in normal tissues (Fig 7F). KIF23 was highly expressed in both normal liver tissues and tumor tissues (Fig 7G). LDLR and TXNIP were observed at different levels of down-regulated in HCC tumor tissues (Fig 7H and 7I).

## CIBERSORT analysis of tumor-infiltrating lymphocytes

The CIBERSORT was used to deconvolute the relative composition of immune cells in the normal and tumor sample from the liver tissue. The bar plot shows the composition ratio of 22 immune cell types in all samples with CIBERSORT P < 0.05 (Fig 8A). The core plot shows the correlation among immune cell subpopulations (Fig 8B), and from the figure, it can be seen that Eosinophils and Dendritic cells activated have a significant positive correlation (correlation coefficient r = 0.97). We compared the infiltration of different immune cells in the tumor and normal liver tissue and found that tumor tissues have higher T cell regulatory (Tregs) infiltration, and the difference is statistically significant (P = 0.018) (Fig 9A). Next, the 4 key mRNAs selected were used for correlation analysis with different types of immune cells, and only the results with P < 0.05 and |r| > 0.4 were listed according to Pearson's correlation coefficient analysis. The analysis showed that the expression of RRM2 has a strong positive correlation with T cells CD4 memory activated, T cells CD8, and T cells follicular helper, and a significant negative correlation with T cells CD4 memory resting (Fig 9B–9E). The expression of TXNIP is closely related to B cell memory (Fig 9F), and there is no immune cell significantly related to KIF23 and LDLR.

## Discussion

Consistently, most HCC patients are diagnosed as advanced HCC at the initial diagnosis, which leads to poor prognosis and higher mortality of HCC worldwide [30]. In terms of

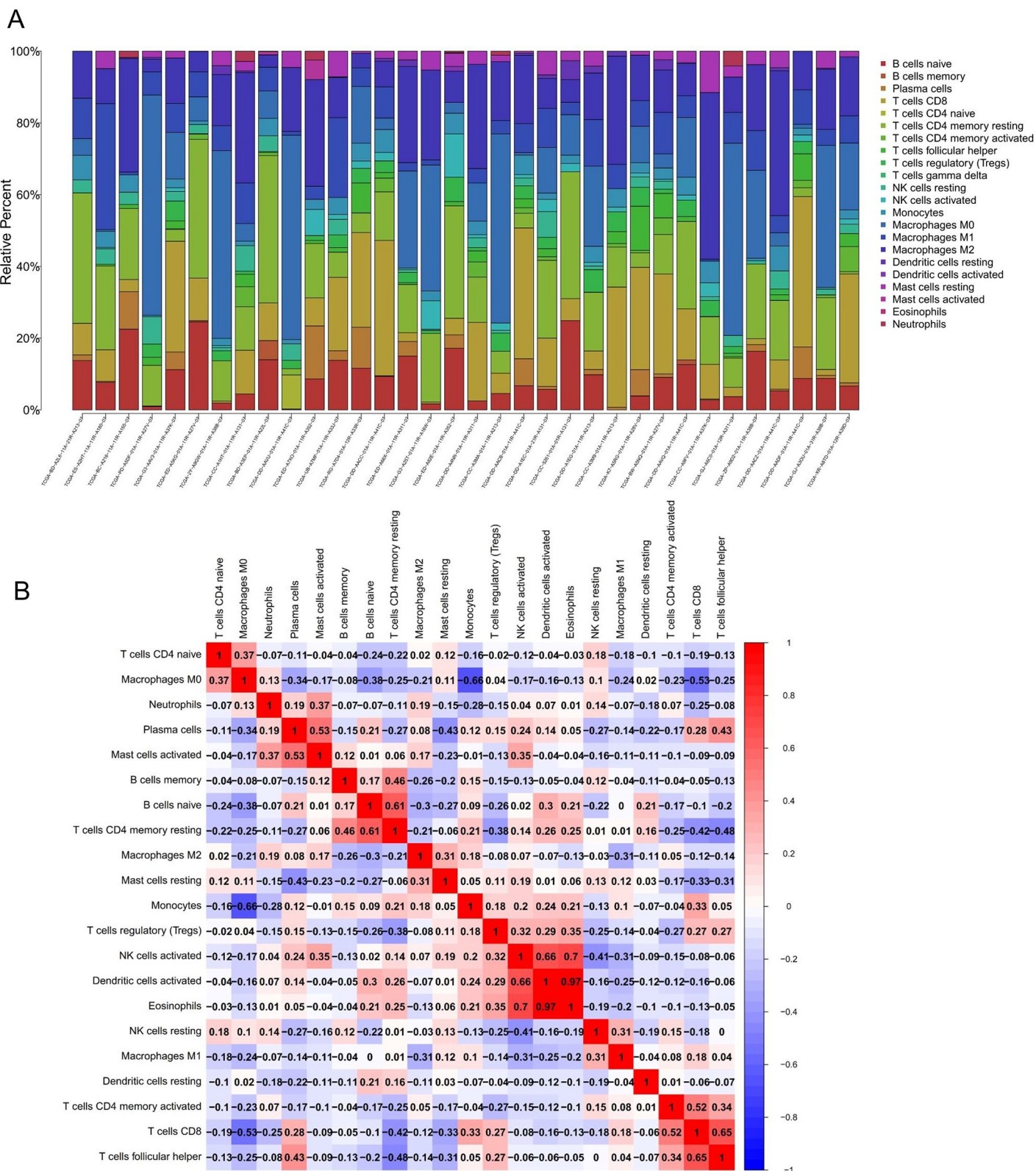

**Fig 8. The immune-cell infiltration was assessed by CIBERSORT algorithm in hepatocellular carcinoma.** (A) The composition ratio of 22 immune cell types. (B) Correlation matrix between the 22 immune cell types.

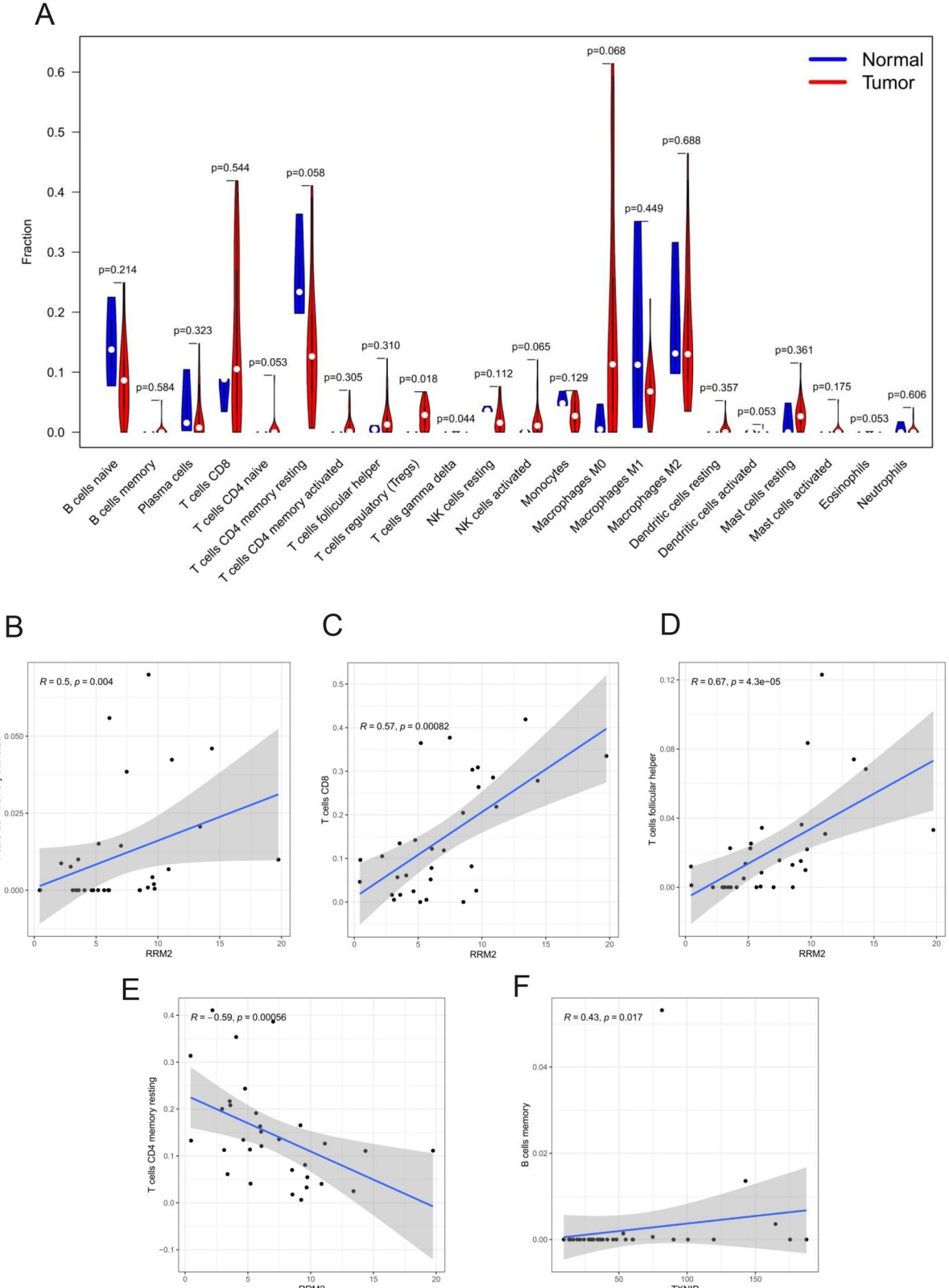

**Fig 9. Correlation between HCC prognosis ceRNA network and immune cells.** (A) Different proportions of immune cell subsets between normal and HCC liver tissues. Pearson correlations indicate that the expression of RRM2 has a positive correlation with T cells CD4

memory activated (B), T cells CD8 (C), and T cells follicular helper (D), and a negative correlation with T cells CD4 memory resting (E). The expression of TXNIP has a positive correlation with B cell memory (F).

treatment methods, cancer immunotherapy has become one of the most promising treatment methods for various advanced solid tumors, including HCC [31]. Immunotherapy has been widely used in the treatment of HCC and has shown remarkable efficacy. However, for some patients with HCC, immunotherapy is ineffective, which may be related to the unique micro-environment of the liver, because the liver is a special organ that induces immune tolerance rather than immunity [32]. This unique and highly heterogeneous immune composition may affect all aspects of HCC, so the implementation of immunotherapy requires a comprehensive understanding of HCC TILs. At present, the regulatory role of ceRNA in malignant tumors has been widely recognized, but the specific mechanism of regulating TILs to affect HCC remains to be determined. LncRNA acts as a ceRNA to target PCED1B mediated by miR-194-5p to promote glioma cell proliferation and inhibit cell apoptosis [33]. LncRNA H19 via sponging miR-152-3p up-regulates the expression of BRD4, thereby promoting the malignant behavior of multiple myeloma cells [34]. Therefore, we constructed a ceRNA regulatory network composed of 67 nodes through WGCNA, in which 44 DElncRNAs and 19 DEGs competed with a limited number of 4 DEmiRNAs. LASSO Cox regression analysis identified 4 key mRNAs in the ceRNA network, including RRM2, KiF23, LDLR, and TXNIP, that could effectively predict prognosis. Correlation analysis showed that these molecules were significantly correlated with TILs. Based on this, we speculated that the ceRNA network and related TILs might play a potentially significant role in the future treatment and prognosis prediction of HCC.

CeRNA plays an important regulatory role in a variety of physiological and pathological processes. C2orf48 was proved to be significantly correlated with the prognosis of liver cancer patients [35]. LINC00152 is associated with a poor prognosis of liver cancer. Cell experiments confirmed that LINC00152 may be involved in the regulation of HCC cell proliferation, cell apoptosis, and cell migration/invasion [36]. In addition, LINC00152 was confirmed to sponge miR-139 and relieve the inhibition of the target gene PIK3CA. Anti-miR-139 can reverse the inhibition of cell proliferation and migration/invasion induced by LINC00152 knockdown [37]. In our study, the low expression of miR-139 was associated with a worse prognosis. Other studies also showed that miR-139 was significantly down-regulated in HCC, and Upregulation of miR-139 promoted the apoptosis of HCC cells and inhibited the growth of tumor cells [38]. SNHG3 is up-regulated in liver cancer, silencing SNHG3 may impair tumor progression, and miR-139-5p has been proved to bind with SNHG3 to play a regulatory role [39]. DIO3OS may inhibit malignant biological behaviors by sponging miR-328 in HCC [40]. The deletion of LINC00261 has been observed in a variety of malignant tumors including HCC, breast cancer, and gastric cancer [41]. LINC00261 can inhibit glycolysis and proliferation of cancer cells [42]. FBXL19-AS1 has been reported to promote tumor progression in both cervical cancer and osteosarcoma [43,44]. In colon cancer, LINC00402 competitively binds miR-141 and miR-424 as the ceRNA of PHLPP2, and LINC00402 has also been confirmed to be associated with metastatic melanoma [45,46]. Other OS-related lncRNAs (WARS2-IT1, TPRG1-AS1, and HAO2-IT1) located in the ceRNA network are less studied, and further studies are needed to clarify the functions of these molecules in HCC and other tumors. We performed KEGG pathway analysis on 19 DEGs in the ceRNA network to characterize the biological functions of the ceRNA network. The enrichment results are shown in Fig 4B, which are mainly enriched in Hepatitis B and Cell cycle. Chronic HBV infection has long been considered an important cause of HCC [47]. HBV integrates into the genome of infected hepatocytes, and promotes the continuous imbalance of hepatocytes through chronic inflammatory

damage, affects the DNA repair mechanism and promotes mutation events, and ultimately leads to the malignant transformation of hepatocytes [48]. The results suggest that the ceRNA network may affect the cell cycle through the interaction with cancer-related miRNAs, thereby affecting the transformation from hepatitis to HCC mediated by HBV. In addition, our study developed a risk score model based on 4 mRNAs (RRM2, KIF23, LDLR and TXNIP) in the ceRNA network, and explored the correlation between these signatures and TILs. The model we constructed showed that as RRM2 and KIF23 expression levels were upregulated, patients' risk scores increased, which represented a worse survival outcome. Confirmed by previous research, compared with non-HCC tissues, the expression of RRM2 in HCC is up-regulated, and inhibiting RRM2 can inhibit the proliferation of HCC cells [49]. RRM2 promotes the occurrence of cell resistance by activating the EGFR/AKT pathway, while lncRNA can up-reg-ulate RRM2 by binding to and inhibiting the expression of miR-139-5p, which is consistent with our research [50]. KIF23 is related to the worse prognosis of HCC patients and has a pro-moting effect on the proliferation and chemoresistance of HCC cells [51]. TXNIP has long been recognized as a potential tumor suppressor gene that inhibits cell growth and is involved in stress response, REDOX regulation, and cell proliferation [52,53].

CIEBERSORT analysis showed that compared with the non-tumor tissues, the infiltration of Tregs in HCC tumor tissues increased. Correlation analysis shows that RRM2 and TXNIP are correlated with CD4+ T cells and B lymphocytes, respectively. CD4+ T cells, mainly com-posed of regulatory T cells (Tregs) and CD4+ helper T cells (Th), among which helper T cells can mediate immunity by secreting various cytokines such as IL-4, IL-10, and interferon-γ [54]. Tregs can suppress tumor immune response by releasing immunosuppressive cytokines [55]. CD4 + T cells are protective in the development of hepatocellular carcinoma in mouse models [56]. It has been reported that the degree of tumor infiltration of B cells is associated with the increased survival rate of HCC patients [57]. TXNIP has a potential role in the forma-tion of germinal centers in peripheral lymphoid organs where B lymphocytes divide rapidly [53]. As a low-density lipoprotein (LDL) receptor, LDLR mediates the absorption of choles-terol by cells [58] and regulates extracellular cholesterol concentration. Also, LDLR is the main pathway to mediate the entry of cholesterol-rich lipoproteins into cancer cells. Blocking this pathway would inhibit tumorigenesis of adenocarcinoma cells, increase the sensitivity of tumor cells to chemotherapy drugs, and promote tumor regression [59]. In breast cancer, increased LDLR expression accelerates tumor growth in mouse models [60]. However, the sit-uation may be the opposite in HCC. In the liver, neutrophil-specific microRNA-223 (miR-223) internalizes into hepatocytes through the expression of LDLR, thereby inhibiting the expression of hepatitis and fibrosis genes and improving nonalcoholic steatohepatitis [61]. Therefore, the lack of LDLR expression in the liver may lead to liver steatosis and persistent inflammation, thus promoting the occurrence of HCC. Therefore, due to the special role of the liver in fat metabolism, the biological role of LDLR in HCC may be different from that in other types of tumors. The specific mechanism needs to be confirmed by further studies.

This study mainly discussed the prognostic value of ceRNA network in HCC and its corre-lation with TILs, and constructed a prognostic model with good sensitivity and specificity. The research results suggest that the ceRNA we constructed may be involved in HCC through TILs and can become potential HCC therapeutic targets, but we still need to further study the molec-ular functions involved in this research in the following clinical, cellular, and animal experiments.

## Conclusion

Our research uses WGCNA, combined with survival analysis and CIBERSORT cell composi-tion analysis to identify the core genes in the prognostic-related ceRNA network and build a

prognostic model based on this. The core genes of this prognostic model were subsequently confirmed to have a significant correlation with B lymphocytes and CD4+ T cells infiltrated in HCC tumor tissue. The results suggest that the 4 mRNAs in the model we constructed participate in the regulation of HCC immune cells, which can be used as potential prognostic indicators for HCC patients.

## Supporting information

**S1 File. The data contained in this file is all the data ultimately used for the analysis in this article.**
(RAR)

## Author Contributions

**Conceptualization:** Ying He.

**Data curation:** Ying He, Rui Xu.

**Formal analysis:** Ying He, Rui Xu, Li Peng.

**Methodology:** Ying He, Rui Xu.

**Software:** Ying He, Li Peng.

**Writing – original draft:** Ying He.

**Writing – review & editing:** Ying He, Xiaoyu Hu.

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
