## [Decision Letter · Decision Letter 0]

7 May 2021

PONE-D-21-00596

Tumor infiltrating lymphocytes associated competitive endogenous RNA networks as predictors of outcome in hepatic carcinoma based on WGCNA analysis

PLOS ONE

Dear Dr. Hu,

Thank you for submitting your manuscript to PLOS ONE. After careful consideration, we feel that it has merit but does not fully meet PLOS ONE’s publication criteria as it currently stands. Therefore, we invite you to submit a revised version of the manuscript that addresses the points raised during the review process.

In particular the quality of certain paragraph and the analysis methods (lack GEO analysis). Discussion should be strenghten.

We look forward to receiving your revised manuscript.

Kind regards,

Isabelle Chemin, PhD

Academic Editor

PLOS ONE

Journal Requirements:

Reviewers' comments:

Reviewer's Responses to Questions

**Comments to the Author**

1. Is the manuscript technically sound, and do the data support the conclusions?

Reviewer #1: Partly

2. Has the statistical analysis been performed appropriately and rigorously? 

Reviewer #1: Yes

3. Have the authors made all data underlying the findings in their manuscript fully available?

Reviewer #1: Yes

4. Is the manuscript presented in an intelligible fashion and written in standard English?

Reviewer #1: Yes

5. Review Comments to the Author

Reviewer #1: The authors performed a study on tumor infiltrating lymphocytes associated competitive endogenous RNA networks as predictors of outcome in hepatic carcinoma based on WGCNA analysis. The research approach is somewhat stereotyped, and some major limitations are as follow:

1. In introduction, the paragraph 2 is not well-written, you wrote the methods of your study in this section, and no actual work was done. You should delete the sentences of methods and write more about some researches on ceRNA, TILs, WGCNA and CIBERSORT.

2. You should use more strict standard of the significance thresholds, for example, log2FC>2 and FDR <0.01, ranther than log2FC>1 and FDR <0.05.

3. In line 70, the abbreviation of differentially expressed mRNAs should be DEGs (differentially expressed genes).

4. You use The Human Protein Atlas Database to validate the key mRNAs, why you did not use your own tissue to validate?

5. In line 130, there are some grammar errors in the title: "Validation of the key mRNAs in prognosis ceRNA network"

6. You need to discuss the ceRNA with the following reference: A comprehensive study of construction and analysis of competitive endogenous RNA networks in lung adenocarcinoma. Biochim Biophys Acta Proteins Proteom. 2020 Aug;1868(8):140444. doi: 10.1016/j.bbapap.2020.140444. Epub 2020 May 11. PMID: 32423886.

7. You need to validate your results using GEO database.

6. PLOS authors have the option to publish the peer review history of their article (what does this mean?). If published, this will include your full peer review and any attached files.

Reviewer #1: No

---

## [Author Response · Author response to Decision Letter 0]

24 Jun 2021

Dear Dr. Isabelle Chemin,

We thank you very much for providing us with an opportunity to revise the manuscript. Many thanks to the reviewers for offering us constructive suggestions to help us improve the quality of the paper. Here we submit a newly revised manuscript with the title "Tumor infiltrating lymphocytes associated competitive endogenous RNA networks as predictors of outcome in hepatic carcinoma based on WGCNA analysis". We carefully considered the reviewers' comments, responded to every point raised by the reviewers and made changes based on the reviewers' suggestions. At the same time, we carefully checked the full text and tried to correct mistakes. We have marked all the changes in yellow in the revised manuscript.

We would like to express our great appreciation to you and reviewers for your comments on our paper. Looking forward to hearing from you.

Thanks for all the help.

Best wishes,

Ying He

The following is a point-to-point response to the two reviewers’ comments.

General comments:

Reviewer #1: The authors performed a study on tumor infiltrating lymphocytes associated competitive endogenous RNA networks as predictors of outcome in hepatic carcinoma based on WGCNA analysis. The research approach is somewhat stereotyped, and some major limitations are as follow:

1. In introduction, the paragraph 2 is not well-written, you wrote the methods of your study in this section, and no actual work was done. You should delete the sentences of methods and write more about some researches on ceRNA, TILs, WGCNA and CIBERSORT.

Your opinion is extremely pertinent and inspired us. We rewrote paragraph 2 of the introduction. A larger space has been added to the discussion about the related research of ceRNA and TILs. The discussion about ceRNA and TILs is divided into separate paragraphs, and the relevant research about WGCNA and CIBERSORT has been supplemented, and the argument for using WGCNA and CIBERSORT has been added. In addition, the last sentence in the first paragraph of the introduction has been deleted and modified into other sentences that we think are more appropriate, which have been highlighted in yellow.

2. You should use more strict standard of the significance thresholds, for example, log2FC>2 and FDR <0.01, ranther than log2FC>1 and FDR <0.05.

Yes, using higher fold change and lower FDR as thresholds to screen differentially expressed genes between groups can make the targets have a lower false-positive rate and the results are more convincing. However, as a result of this, certain genes that play an important role in the occurrence and development of the disease but have relatively small differences in expression between tumors and normal tissues may be filtered out. At the same time, giving up a large number of genes that are correct for a lower false discovery rate will make the research less comprehensive. The significance thresholds of log2FC>1 and FDR <0.05 is a very commonly used filter standard. Choosing this filter standard can include more differential genes that may play an important role in the disease process while ensuring a relatively low false-positive rate. In addition, many published studies have used log2FC>1 and FDR <0.05 as a screening criterion for research [1-3]. That is to say, this filtering standard has been proved to be feasible many times. This is why we chose log2FC>1 and FDR <0.05 as the screening criteria for differential genes.

3. In line 70, the abbreviation of differentially expressed mRNAs should be DEGs (differentially expressed genes).

In line 92 of the revised manuscript, we have changed “Differentially expressed mRNAs (DERNAs)” to “Differentially expressed genes (DEGs)”, and made corresponding adjustments to the sentence. Modifications have been made to the entire article involving errors related to “Differentially expressed mRNAs” and “DEmRNAs”.

4. You use The Human Protein Atlas Database to validate the key mRNAs, why you did not use your own tissue to validate?

It is certainly better to use our own tissues to verify the expression of key genes, which can make our research more convincing. We have tried to do this before, but due to lack of sufficient funding and limited experimental time, we have no way to carry out related experiments. This is also the unavoidable shortcoming of our research. We can only look forward to furthering completing the relevant supplementary experiments after obtaining the support of relevant conditions in the future.

The Human Protein Atlas database provides information on the tissue and cell distribution of a variety of human proteins. In this database, researchers use highly specific antibodies, using immunoassay technology (immunohistochemistry, immunoblotting, and

Immunofluorescence), the detailed detection of the expression of each protein in different cell lines, human normal and tumor tissues. And it has been cited by many studies as the only verification tool for the target protein without additional tissue or cell verification [4-8]. It shows that the acceptance and recognition of the database is generally high, which can save the use of clinical specimens and related experimental expenses to a certain extent.

5. In line 130, there are some grammar errors in the title: "Validation of the key mRNAs in prognosis ceRNA network"

In the revised manuscript, in line 154, we have changed "Validation of the key mRNAs in prognosis ceRNA network" to "Prognosis and expression validation for key DEGs".

6. You need to discuss the ceRNA with the following reference: A comprehensive study of construction and analysis of competitive endogenous RNA networks in lung adenocarcinoma. Biochim Biophys Acta Proteins Proteom. 2020 Aug;1868(8):140444. doi: 10.1016/j.bbapap.2020.140444. Epub 2020 May 11. PMID: 32423886.

Thanks to the revised template provided by the reviewers, we are fully aware of our shortcomings in the discussion part, so we have made detailed revisions to the entire discussion, almost rewritten this part, and added a lot of ceRNA discussions and references to related literature.

7. You need to validate your results using GEO database.

We downloaded 5 datasets GSE14520, GSE22058, GSE25097, GSE57957, and GSE76427 from GEO database for expression verification of key DEGs. Data were plotted in GraphPad Prism and statistical analyses were performed with SPASS. Since GSE14520 contains a large number of cases and has complete survival data, we used GSE14520 as the validation set to verify the sensitivity and accuracy of the risk score model we built. At the same time, because the oncomine database integrates data from GEO, TCGA, and other sources, the data in the oncomine database are partially duplicated in TCGA and GEO. We have added GEO for relating verification. Therefore, we have deleted the previous verification results obtained using the oncomine database.

The manuscript has been modified in detail following the requirements listed in "PLOS Manuscript Body Formatting Guidelines" and "PLOSOne_formatting_sample_title_authors_affiliations" to ensure that our manuscript meets PLOS ONE's style requirements.

Although some issues were not pointed out by the reviewers, we believe that changes need to be made. The details are as follows:

#1 The WGCNA analysis was used in this study, and the final mRNAs used for LASSO Cox analysis were all the mRNAs intersected with the core module mRNAs screened by WGCNA. Therefore, the final construction of the ceRNAs network should be incorporated into the analysis results of WGCNA, and the construction should be based on the final 19 intersection mRNAs. Therefore, we narrowed down the originally constructed ceRNAs regulatory network and selected 19 intersection mRNAs to reconstruct a ceRNAs network containing 44 DElncRNAs, 19 DEGs, and 4 DEmiRNAs. The order of the subsections of results has been adjusted, and the subsection "Construction of WGCNA and identification of hub modules" has been replaced in front of "Construction of the ceRNA network". That is, WGCNA analysis is performed first, and then the construction of ceRNAs. 

#2 In addition to using GEO database to verify the key DEGs verification, we also performed a difference analysis on GSE14520. The mRNAs finally used for LASSO Cox analysis are no longer the previous 19 mRNAs but 14 mRNAs. These 14 mRNAs are differentially expressed in GSE14520 and have the same changing trend. In the revised manuscript, the relevant parts involved have been revised accordingly.

#3 Since the analysis results of the GEO database were added, the signatures of the risk score model finally constructed changed. As a result, the previous KPNA2, LDLR, TXNIP have become LDLR, TXNIP, RRM2, KIF23, and the corresponding coefficients in the model have also changed.

#4 In the process of careful revision of the manuscript, we believed that the failure to carry out survival analysis on the entire ceRNA we constructed was a shortcoming that needed to be remedied in time. Although the reviewer did not raise this issue, we believed that survival analysis on the ceRNA network must be carried out to better demonstrate the prognostic value of this ceRNA. Therefore, we have supplemented this part, and the analysis results of the added part are shown in Fig.6.

Reference

1. Wang Z, Song M, Li Y, Chen S, Ma H. Differential color development and response to light deprivation of fig (Ficus carica L.) syconia peel and female flower tissues: transcriptome elucidation. BMC Plant Biol [Internet]. 2019 May 23 [cited 2021 Jun 5];19. Available from: https://www.ncbi.nlm.nih.gov/pmc/articles/PMC6533723/

2. Ahmed AA, Marchetti C, Ohnmacht SA, Neidle S. A G-quadruplex-binding compound shows potent activity in human gemcitabine-resistant pancreatic cancer cells. Sci Rep [Internet]. 2020 Jul 22 [cited 2021 Jun 5];10. Available from: https://www.ncbi.nlm.nih.gov/pmc/articles/PMC7376204/

3. Safronov O, Kreuzwieser J, Haberer G, Alyousif MS, Schulze W, Al-Harbi N, et al. Detecting early signs of heat and drought stress in Phoenix dactylifera (date palm). PLoS One [Internet]. 2017 Jun 1 [cited 2021 Jun 5];12(6). Available from: https://www.ncbi.nlm.nih.gov/pmc/articles/PMC5453443/

4. Hua X, Chen J, Ge S, Xiao H, Zhang L, Liang C. Integrated analysis of the functions of RNA binding proteins in clear cell renal cell carcinoma. Genomics. 2021 Jan;113(1 Pt 2):850–60.

5. Ding H, Xiong X-X, Fan G-L, Yi Y-X, Chen Y-R, Wang J-T, et al. The New Biomarker for Cervical Squamous Cell Carcinoma and Endocervical Adenocarcinoma (CESC) Based on Public Database Mining. Biomed Res Int [Internet]. 2020 Apr 12 [cited 2021 Jun 5];2020. Available from: https://www.ncbi.nlm.nih.gov/pmc/articles/PMC7174939/

6. Sun Y, Dai W-R, Xia N. Comprehensive analysis of lncRNA-mediated ceRNA network in papillary thyroid cancer. Eur Rev Med Pharmacol Sci. 2020 Oct;24(19):10003–14.

7. Li H, Liu J, Shen S, Dai D, Cheng S, Dong X, et al. Pan‐cancer analysis of alternative splicing regulator heterogeneous nuclear ribonucleoproteins (hnRNPs) family and their prognostic potential. J Cell Mol Med. 2020 Oct;24(19):11111–9.

8. Zeng Z, Cao Z, Tang Y. Increased E2F2 predicts poor prognosis in patients with HCC based on TCGA data. BMC Cancer [Internet]. 2020 Oct 28 [cited 2021 Jun 5];20. Available from: https://www.ncbi.nlm.nih.gov/pmc/articles/PMC7594443/

---

## [Decision Letter · Decision Letter 1]

6 Jul 2021

Tumor infiltrating lymphocytes associated competitive endogenous RNA networks as predictors of outcome in hepatic carcinoma based on WGCNA analysis

PONE-D-21-00596R1

Dear Dr. Hu,

We’re pleased to inform you that your manuscript has been judged scientifically suitable for publication and will be formally accepted for publication once it meets all outstanding technical requirements.

Kind regards,

Isabelle Chemin, PhD

Academic Editor

PLOS ONE

Additional Editor Comments (optional):

Reviewers' comments:

Reviewer's Responses to Questions

**Comments to the Author**

1. If the authors have adequately addressed your comments raised in a previous round of review and you feel that this manuscript is now acceptable for publication, you may indicate that here to bypass the “Comments to the Author” section, enter your conflict of interest statement in the “Confidential to Editor” section, and submit your "Accept" recommendation.

Reviewer #1: All comments have been addressed

2. Is the manuscript technically sound, and do the data support the conclusions?

Reviewer #1: Yes

3. Has the statistical analysis been performed appropriately and rigorously? 

Reviewer #1: Yes

4. Have the authors made all data underlying the findings in their manuscript fully available?

Reviewer #1: Yes

5. Is the manuscript presented in an intelligible fashion and written in standard English?

Reviewer #1: Yes

6. Review Comments to the Author

Reviewer #1: (No Response)

7. PLOS authors have the option to publish the peer review history of their article (what does this mean?). If published, this will include your full peer review and any attached files.

Reviewer #1: No

---

## [Editor Report · Acceptance letter]

21 Jul 2021

PONE-D-21-00596R1 

Tumor infiltrating lymphocytes associated competitive endogenous RNA networks as predictors of outcome in hepatic carcinoma based on WGCNA analysis 

Dear Dr. Hu:

I'm pleased to inform you that your manuscript has been deemed suitable for publication in PLOS ONE. Congratulations! Your manuscript is now with our production department. 

Kind regards, 

on behalf of

Mrs Isabelle Chemin 

Academic Editor

PLOS ONE